# Structural basis of ion transport and inhibition in ferroportin

Yaping Pan [1,4 ✉], Zhenning Ren[1,4], Shuai Gao [2,4 ✉], Jiemin Shen [1,4], Lie Wang[1], Zhichun Xu[1], Ye Yu[1], Preetham Bachina[1], Hanzhi Zhang[1], Xiao Fan[2], Arthur Laganowsky [3], Nieng Yan [2] & Ming Zhou [1 ✉]

Ferroportin is an iron exporter essential for releasing cellular iron into circulation. Ferroportin is inhibited by a peptide hormone, hepcidin. In humans, mutations in ferroportin lead to ferroportin diseases that are often associated with accumulation of iron in macrophages and symptoms of iron deficiency anemia. Here we present the structures of the ferroportin from the primate Philippine tarsier (TsFpn) in the presence and absence of hepcidin solved by cryo-electron microscopy. TsFpn is composed of two domains resembling a clamshell and the structure defines two metal ion binding sites, one in each domain. Both structures are in an outward-facing conformation, and hepcidin binds between the two domains and reaches one of the ion binding sites. Functional studies show that TsFpn is an electroneutral $H^+/Fe^{2+}$ antiporter so that transport of each $Fe^{2+}$ is coupled to transport of two $H^+$ in the opposite direction. Perturbing either of the ion binding sites compromises the coupled transport of $H^+$ and $Fe^{2+}$. These results establish the structural basis of metal ion binding, transport and inhibition in ferroportin and provide a blueprint for targeting ferroportin in pharmacological intervention of ferroportin diseases.

[1] Verna and Marrs McLean Department of Biochemistry and Molecular Biology, Baylor College of Medicine, Houston, TX 77030, USA. [2] Department of Molecular Biology, Princeton University, Princeton, NJ 08544, USA. [3] Department of Chemistry, Texas A & M University, College Station, TX 77843, USA. [4]These authors contributed equally: Yaping Pan, Zhenning Ren, Shuai Gao, Jiemin Shen. ✉email: yaping.pan@bcm.edu; shuaig@princeton.edu; mzhou@bcm.edu

In mammals, ferroportin (Fpn) is highly expressed in enterocytes, hepatocytes, and macrophages to export iron ions that are absorbed from food or recovered from digestion of senescent red blood cells[1]. Fpn also mediates iron transport across the placenta and thus is required for the normal development of embryos[2]. Fpn activity is regulated by hepcidin, a peptide hormone, which is known to reduce Fpn activity by a combination of inhibiting the transport activity and promoting endocytosis of Fpn[3,4]. More than 60 mutations in Fpn are associated with ferroportin diseases in human[5,6], indicating an important physiological role of Fpn in iron homeostasis. Fpn and its modulation by hepcidin has been the focus of targeted therapeutics for treating ferroportin diseases[7–9].

Fpn belongs to the solute carrier family 40 (SLC40A1)[10–12] and is a member of the major facilitator superfamily (MFS) of secondary transporters which includes glucose transporters (GLUT or SLC2A)[13–15], peptide transporters (PEPT or SLC15A)[16–18], and equilibrative nucleoside transporters (ENT or SLC29A)[19]. Transporters of the MFS family share a common structural fold that has two homologous halves forming a clamshell like architecture. A single substrate binding site is commonly located to the center of the clamshell, and substrate translocation is achieved by rocker-switch type motions of the two halves of the clamshell so that the substrate binding site is alternatively exposed to either side of the membrane[20]. Structures of a bacterial homolog of Fpn (*Bdellovibrio bacteriovorous*; BbFpn) in both the inward- and outward-facing conformations were reported recently[21,22], which significantly enhances our understanding of Fpn. However, given the modest sequence identity (~20%) and similarity (49%) between BbFpn and human Fpn, structures of mammalian Fpn are required to address questions on interactions with hepcidin and binding and transport of metal ions.

In this work, we expressed and purified Fpns from human and from Philippine tarsier (*Tarcius syrichta* or *Carlito syrichta*; TsFpn), which is 92% identical and 98% similar to human Fpn, and we demonstrated that Fpn is an electroneutral $H^+/Fe^{2+}$ antiporter. We also determined structures of TsFpn in the presence of $Co^{2+}$ or hepcidin, and the structures define two metal ion binding sites as well as the hepcidin binding pocket.

## Results

**In vitro functional studies of TsFpn.** Purified TsFpn elutes as a single peak on a size-exclusion chromatography column, and the elution volume is consistent with TsFpn being a monomer (Supplementary Fig. 1a–c and Methods section). We reconstituted the purified TsFpn protein into liposomes and estimated uptake of $Fe^{2+}$ by following the fluorescence of calcein trapped inside of the liposomes (Fig. 1a and Methods section). When $Fe^{2+}$ (100 μM) was added to the external solution, rapid quench of internal calcein fluorescence was observed in liposomes reconstituted with TsFpn and the quench reached ~35% after 400 s while slow and modest change of fluorescence was observed in liposomes without TsFpn and the quench reached <7% at the end of 400 s (Fig. 1b). We then measured the time course of fluorescence quench at different concentrations of $Fe^{2+}$ (Fig. 1b), and the data can be fit with a two component exponential decay function with a fast initial phase and a slower later phase (Methods). The initial rate of fluorescence quench, which we approximate as the initial rate of metal ion transport, allows comparison of transport in different concentrations of $Fe^{2+}$ (Fig. 1c). The data can be fit to a Michaelis-Menten (MM) equation with a $K_M$ of 1.9 ± 0.26 μM and $V_{max}$ of 0.13 ± 0.09 $min^{-1}$ (Fig. 1c). Since $Fe^{2+}$ is easily oxidized under aerobic conditions, a reducing reagent (1 mM vitamin C) was included in the external solution to stabilize $Fe^{2+}$. To eliminate complications from oxidation of $Fe^{2+}$, we

tested transport of $Co^{2+}$, which is more stable than $Fe^{2+}$ (Fig. 1d). The initial rate of $Co^{2+}$ uptake versus $Co^{2+}$ concentrations can also be fit with a Michaelis–Menten equation with a $K_M$ of 8.7 ± 2.4 μM and $V_{max}$ of 0.12 ± 0.01 $min^{-1}$ (Fig. 1e). These results indicate that TsFpn transports $Fe^{2+}$ and $Co^{2+}$ similarly and thus we will use $Co^{2+}$ as a surrogate for $Fe^{2+}$ in subsequent measurements. Although $K_M$ and $V_{max}$ values allow comparison of $Fe^{2+}$ and $Co^{2+}$ transport in Fpn, we emphasize that the mechanism of metal ion transport in Fpn is more complicated than a simple two-step enzymatic reaction from which the MM equation was derived.

Because extracellular $Ca^{2+}$ was shown to be required for iron transport in human Fpn[22] and since our in vitro transport assay has no Ca2+ present, we next measured $Co^{2+}$ uptake in TsFpn in the presence of 2 mM $Ca^{2+}$ added on the external side (Fig. 1f–h). As we will show later, TsFpn assumes both orientations when reconstituted into proteoliposomes so that some of the transporters have the extracellular side exposed to the external side of the liposomes. As shown in Fig. 1f, h, metal ion transport by TsFpn is not significantly different in the presence or absence of $Ca^{2+}$. We also purified human Fpn and measured $Co^{2+}$ transport (Fig. 1g and Supplementary Fig. 1d). Human Fpn transported $Co^{2+}$ with a slightly slower initial rate than that of TsFpn, and the rate of uptake was not affected by $Ca^{2+}$ (Supplementary Table 2). These results indicate that $Ca^{2+}$ is not required for Fpn function in the in vitro flux assay. However, there are technical limitations of the in vitro assay, for example, the presence of $Co^{2+}$ on the same side of $Ca^{2+}$ could potentially mask the effect of $Ca^{2+}$ potentiation. Thus, more studies are required to determine the role of $Ca^{2+}$ on behaviors of human Fpn protein.

We next measured $Co^{2+}$ binding to the purified TsFpn using isothermal titration calorimetry (ITC) and found that binding of $Co^{2+}$ is exothermic with a $\Delta H$ of −12.0 ± 0.55 kJ/mol and TΔS of 9.29 ± 0.38 kJ/mol, and an estimated dissociation constant ($K_d$) of 182.6 ± 16.8 μM (Supplementary Fig. 2a). The modest affinity seems consistent with Fpn as an exporter in cells in which iron ions are abundant. However, factors such as solubilization of Fpn in detergent and heterogeneity in conformations of the Fpn could influence metal ion affinity. Such complications notwithstanding, the ITC measurement provides a description of Fpn property independent of the transport assay. It has not escaped our notice that $K_M$ is ~20 fold smaller than $K_d$. For an enzymatic reaction with a simple kinetic mechanism composed of substrate binding and then conversion to a product, $K_M$ is always larger than $K_d$ (Methods section). The reversal of $K_M$ and $K_d$ is an indication that Fpn undergoes a more complex kinetic mechanism to transport metal ions. However, we do not have enough data to lay out the full kinetic mechanism of transport. In addition, $K_d$ was measured on TsFpn solubilized in detergent while $K_M$ on TsFpn in proteoliposomes. Interestingly, a similar phenomenon was reported for a mitochondrial iron exporter in which the $K_M$ is 4 μM and $K_d$ is 450 μM[23].

We developed mouse monoclonal antibodies against TsFpn to facilitate its structural determination[24] (Methods section). TsFpn forms a stable complex with the antigen binding fragment (Fab) of a monoclonal antibody 11F9 as indicated by a shorter retention time on the size-exclusion column (Supplementary Fig. 1a, b). To assess the effect of 11F9 Fab on TsFpn, we examined $Co^{2+}$ binding and transport in the presence of the Fab. TsFpn-11F9 Fab complex has a modestly reduced affinity to $Co^{2+}$ with a $K_d$ of 258.2 ± 34.2 μM (Supplementary Fig. 2b). Interestingly, 11F9 Fab inhibits $Co^{2+}$ uptake, and the rate of uptake is reduced by ~50% when the Fab (20 μM) is added to the external side of the liposomes and ~85% when the Fab is included on both sides of the liposomes (Fig. 1i–l and Supplementary Table 2). Since Fab binds to the intracellular side the Fpn (see below), this result indicates that TsFpn is reconstituted into the liposomes with both

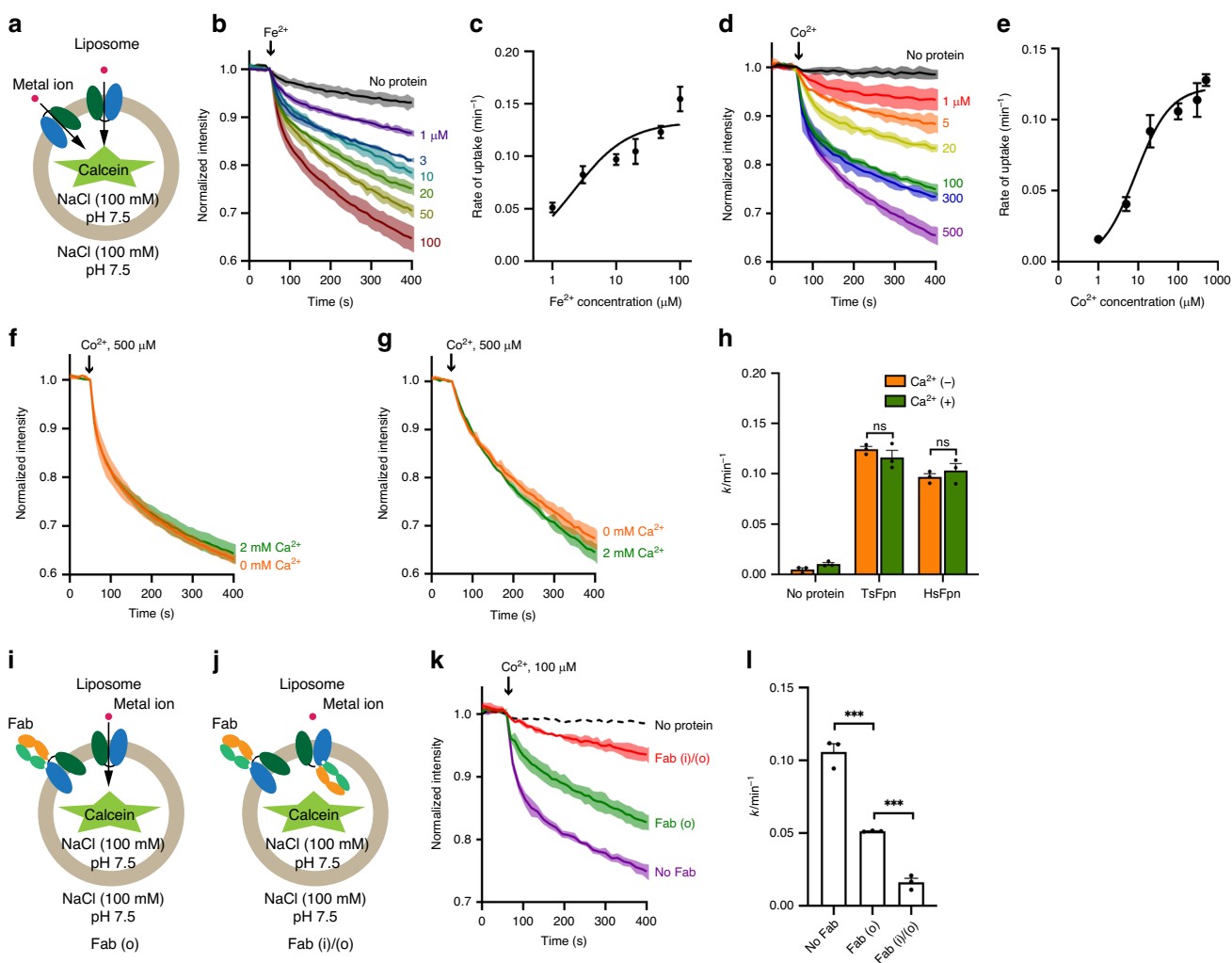

**Fig. 1 Function of purified TsFpn. a** Schematic view of a TsFpn-containing proteoliposome with a membrane impermeable calcein dye enclosed. Metal ions ($Fe^{2+}$ or $Co^{2+}$, red spheres) are added to the external side and uptake of the metal ions quenches the fluorescence. Two orientations of TsFpn are shown. **b** Quench of calcein fluorescence in the presence of indicated concentrations of external $Fe^{2+}$ for liposomes with TsFpn (colored traces) and without TsFpn (black trace, 100 μM $Fe^{2+}$). **c** The initial rate of fluorescence quench versus concentrations of $Fe^{2+}$. The solid curve is data fit to a Michaelis–Menten equation. **d** Quench of calcein fluorescence in the presence of indicated concentrations of external $Co^{2+}$ for liposomes with TsFpn (colored traces) and without TsFpn (black trace, 500 μM $Co^{2+}$). **e** The initial rate of fluorescence quench versus concentrations of $Co^{2+}$. The solid curve is data fit to a Michaelis–Menten equation. **f–h** $Co^{2+}$ transport in the presence and absence of 2 mM $Ca^{2+}$ by TsFpn (**f**) or human Fpn (**g**). Initial rate constants of uptake in TsFpn and HsFpn plotted as bar graphs (**h**). Two-way ANOVA: within each group, no significant difference (TsFpn: $p = 0.54$; HsFpn: $p = 0.70$). **i, j** Schematic view of orientation-specific binding of Fab to TsFpn in proteoliposomes when Fab is added to the outside (**i**) or to both sides (**j**) of proteoliposomes. **k** $Co^{2+}$ transport in the absence of Fab (purple trace), in the presence of Fab on the outside (green trace) and on both sides (red trace). **l** Initial rate constants of transport for traces shown in **k**. Two-tailed Student's $t$ test: ***$p < 0.001$. Throughout this article, all time-dependent fluorescence traces are shown as solid lines with a shade in which the solid line represents the mean value of at least three independent experiments and the width of the shade represents the standard deviation; For all bar graphs, a scatter plot is overlaid on each bar and the height is the mean of at least three measurements and the error bar is the standard error of the mean (s.e.m.).

orientations. Combined, we conclude that 11F9 Fab forms a stable complex with TsFpn and has a modest effect on ion binding and a pronounced inhibitory effect on ion transport.

**Structure of TsFpn**. We reconstituted the TsFpn-11F9 Fab complex into nanodiscs (Supplementary Fig. 1a) and prepared cryo-EM grids in the presence of 10 mM of $Co^{2+}$. The images show recognizable particles of TsFpn-11F9 Fab complex and we obtained a final map at 3.0 Å overall resolution (Fig. 2a, Supplementary Figs. 3a–e and 4a, and Supplementary Table 1). The map shows clear density for all transmembrane helices and resolves most of the side chains (Supplementary Fig. 5a) to allow de novo building of the TsFpn structure. The final structure

model includes residues 17 to 237, 289–395 and 453–552. The N-terminal 16 residues, two loops between TM6 and 7 and TM9 and 10, and C-terminal 25 residues were not resolved (Fig. 2c) and these regions are predicted to be unstructured (Supplementary Fig. 6). For the Fab fragment, although the constant region was not fully resolved and built as polyalanines, the variable region was well-resolved with a local resolution close to 2.9 Å (Supplementary Fig. 3).

TsFpn adopts a canonical MFS fold[25]. The 12 transmembrane helices are packed into two well-defined domains. TM1-6 form the N-domain, and TM7-12 the C-domain (Fig. 2b–d). Based on previous studies of human Fpn topology[26] and the "positive-inside" rule[27], both the N- and C-termini are located to the

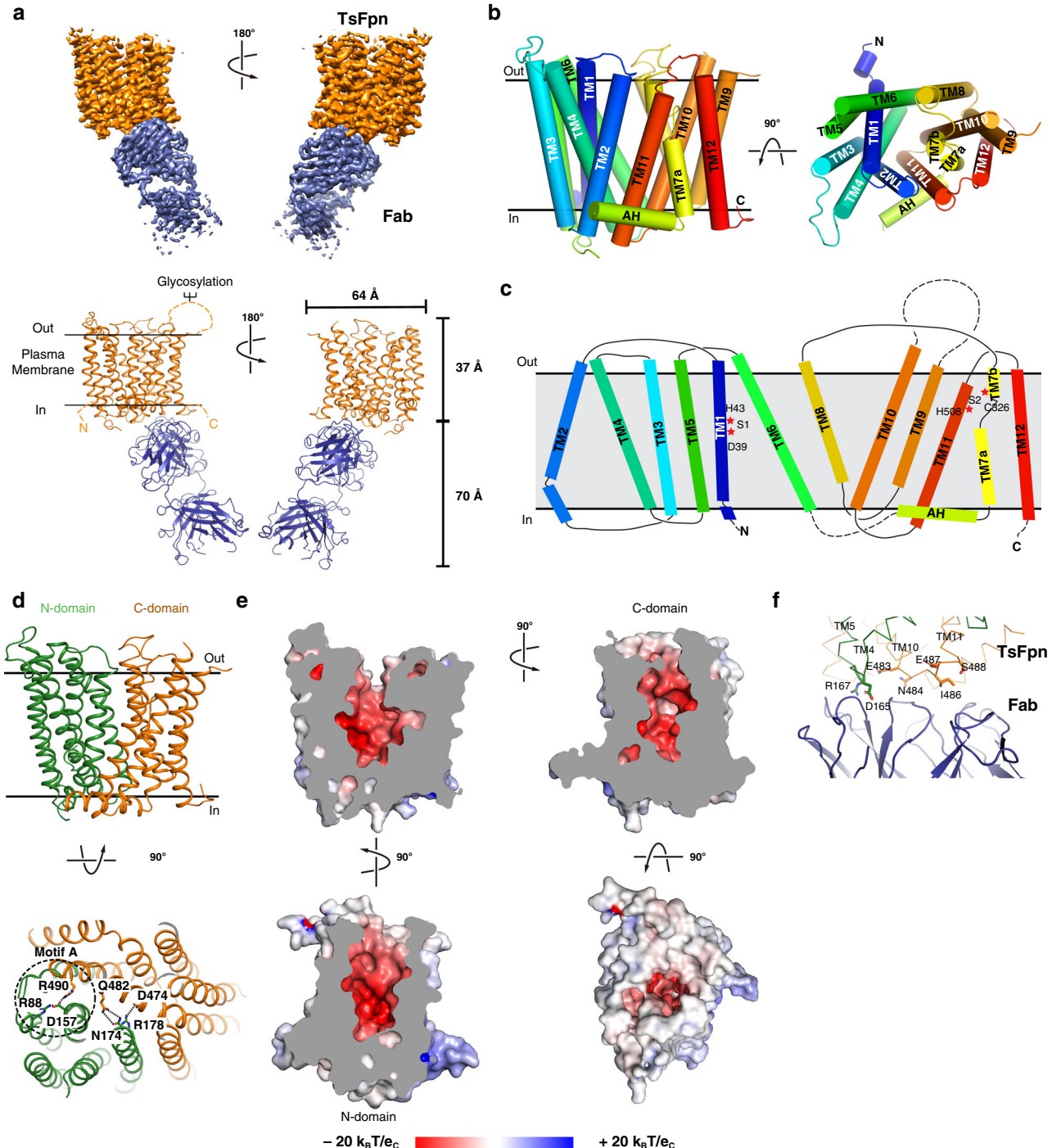

**Fig. 2 Structure of TsFpn. a** Top panel: cryo-EM map of TsFpn (orange) in complex with Fab (blue) in two views. Bottom panel: TsFpn in complex with Fab shown as ribbon representation in two views. **b** TsFpn structure shown as cylinder representation in two views. **c** Topology of TsFpn. Regions that are not resolved in the structure are marked as dotted lines. **d** Top: The N- and C-domains of TsFpn shown in green and orange, respectively. Bottom: TsFpn viewed from the intracellular side. Interacting residues from the N- and C-domains are marked as sticks. **e** Electrostatic potential of TsFpn mapped onto the surface representation. The cut-away views show the large cavity formed between the N- and C- domains. Electrostatic potential is calculated by APBS[55] in Pymol. **f** Interface between TsFpn (green and orange ribbons) and Fab (blue cartoon). Residues from the loop between TM10-11 are shown as orange sticks, and the two charged residues from the N-domain are shown as green sticks.

cytosolic side. The 11F9 Fab binds to the intracellular side of TsFpn. The N- and C-domains are connected by a long intracellular loop between TM6 and TM7. Part of the loop is an amphipathic helix (AH) that extends horizontally and is oriented parallel to the intracellular surface of the membrane (Fig. 2b). A distinctive feature of TsFpn is that its TM7 is

composed of two helices, TM7a and TM7b, which are connected by a loop (residues 319–323). This structural feature was also present in the bacterial homolog BbFpn. Structure of TsFpn aligns well with that of BbFpn in the outward-facing conformation (PDB ID 5AYN) and the root mean squared distance of Cα is 1.39 Å (Supplementary Fig. 7a). Although TM7a aligns quite well

between the two structures, TM7b shows a pronounced shift (Supplementary Fig. 7b).

The N- and C-domains make contact at the cytosolic side. Asp157 on TM4 is close enough to both Arg88 on TM3 and Arg490 on TM11 to form salt bridges (Fig. 2d). The three charged residues are commonly known as motif-A[25] in the MFS family of transporters, which is composed of conserved charged residues and the electrostatic interactions are thought to stabilize the outward-facing conformation. Other interactions between the N- and C-domains include Arg178 in the N-domain and Asp474 in the C-domain, and Asn174 in the N-domain and Gln482 in the C-domain (Fig. 2d). All the interacting pairs in TsFpn are conserved in human Fpn (Supplementary Fig. 6) and mutations to Arg88, Asp157, Asn174, Arg178, and Arg490 are known to cause ferroportin diseases[1,5,28]. The variable region of the 11F9 Fab interacts with a short intracellular loop between TM10 and 11 of the C-domain, and with residues Asp165 and Arg167 from the N-domain (Fig. 2f). Because 11F9 Fab bridges both domains when the transporter is in the outward-facing conformation, we speculate that the Fab reduces metal ion transport by stabilizing the outward-facing conformation of TsFpn.

**Metal ion binding sites**. TsFpn structure has a large solvent-accessible central cavity between the N- and C-domains (Fig. 2e). Residues lining the cavity are mostly charged or hydrophilic, and although there are several arginine residues, the overall electrostatic surface potential of the cavity is highly negative (Fig. 2e). Two strong non-protein densities stand out in the cavity and we assigned the two as $Co^{2+}$ binding sites. The first site, S1, is in the N-domain and coordinated by Asp39 and His43 from TM1, and the second site, S2, is in the C-domain and coordinated by Cys326 from TM7b and His508 from TM11 (Fig. 3a–c). The density at S1 has higher signal level ($12.5\sigma$) than that at S2 ($9\sigma$) and both densities have higher signal levels than those of the surrounding residues (Fig. 3b, c). Both S1 and S2 are solvent accessible from the extracellular side, and the distance between the two sites is 16.0 Å as measured between the two ions. S1 and S2 are both coordinated by two residues, which is different from ion binding sites identified in other transition metal ion transporters of known structures, such as NRAMP[29,30], VIT1[31], YiiP[32], ZIP[33], and ZneA[34], all of which have at least four residues coordinating a metal ion binding site. Although S2 does not have a charged residue making direct contact with the bound ion, two negatively charge residues, Asp325 and Asp505 are located close to S2 and could interact with Cys326 and His508, respectively. It seems unusual to have two substrate binding sites because most other members of the MFS family of transporters have a single substrate binding site often coordinated by residues from both the N- and C-domains[13–19].

Although the overall architectures of TsFpn and BbFpn are similar, locations of the ion binding sites differ in the two structures. In the initial BbFpn structure (PDB ID 5AYN), an iron ion binding site was identified to be coordinated by residues Thr20 and Asp24 on TM1 and Asn196, Ser199, and Phe200 on TM6. These residues are well conserved in TsFpn, Ser35, and Asp39 on TM1 and Asn212, Ser215, and Met216 on TM6. However, these residues do not form an ion binding site in TsFpn because TM6 is farther away from TM1 and Asp39 has a different rotamer conformation (Supplementary Fig. 7a, c, d). In a more recent publication, BbFpn structure was solved in an inward-facing conformation (PDB ID 6BTX) with a $Ca^{2+}$ binding site coordinated by residues Asp24 from TM1, Gln84 from TM3, and Asn196 and Glu203 from TM6[22]. Although these residues are conserved in TsFpn, they are also not in position to form an ion binding site (Supplementary Fig. 7g, h). In the recent BbFpn

structure, a $Ni^{2+}$ ion binding site was identified and it is coordinated by His261 and an EDTA molecule[22]. His261 in BbFpn corresponds to Asp325 in TsFpn, which is close to S2 in TsFpn (Supplementary Fig. 7e, f). Although the precise location and coordination of the ion binding sites are different in TsFpn and BbFpn, the two share the features of having two metal ion binding sites, one in the N-domain close to TM1 and another in the C-domain close to the loop between TM7a and 7b.

**TsFpn is an electroneutral $H^+/Fe^{2+}$ antiporter**. As a first test to validate the ion binding sites, we examined pH dependency of metal ion binding in TsFpn because both the S1 and S2 sites have a histidine. TsFpn does not bind to $Co^{2+}$ in pH 6.0, and the binding affinity gradually recovers as pH increases from 6.0 to 8.0 (Supplementary Fig. 2a, c–g). These results suggest that pH could play a role in metal ion transport. Indeed, pH-dependent transport was reported for human Fpn in a cell-based assay[35], in which a bell-shaped curve was obtained: iron ion export is the highest at pH 7.5, and lower pH (5.0–6.5) inactivates the transport while higher pH (8.0 and 8.5) slightly reduces the transport. In BbFpn, $Co^{2+}$ transport is enhanced by a higher pH[21].

To examine the pH effect on metal ion transport in TsFpn, we started by keeping the pH at 7.5 inside of liposomes and varying the pH outside from 6.5 to 8.5. The rate of $Co^{2+}$ uptake is the slowest when external pH is 6.5 and the rate increases as external pH increases from 6.5 to 8.5 (Fig. 4a and Supplementary Table 2). We noticed that the rate of $Co^{2+}$ transport under a pH gradient, pH 7.5 (in)/8.5(out), $0.54 \pm 0.037$ min$^{-1}$, is significantly higher than that in symmetrical pH 7.5 (Fig. 4b). We wondered whether a $H^+$ gradient is responsible for the enhanced metal ion transport. When the pH gradient is eliminated with symmetrical pH 8.5 inside and outside of the liposomes, the rate of $Co^{2+}$ transport is reduced to $0.20 \pm 0.0057$ min$^{-1}$ (Fig. 4a, b). This result is consistent with the hypothesis that pH gradient enhances the rate of $Co^{2+}$ uptake, i.e., a higher $H^+$ concentration inside of the liposomes facilitates metal ion uptake. We further tested this hypothesis in the following three experiments.

We first varied salt composition in the assay buffer and we found that the rate of $Co^{2+}$ transport is not significantly different when the initial condition of KCl inside and NaCl outside of the vesicles was switched to symmetrical NaCl, KCl, or K-Gluconate, indicating that neither $Na^+$, $K^+$, or $Cl^-$ has a significant effect on $Co^{2+}$ transport (Fig. 4c, d). Second, we visualized proton transport by monitoring the quench of fluorescence of a proton-sensitive dye pyranine trapped inside of the liposomes (Methods). In this experiment, $Co^{2+}$ concentration was ~2 mM inside and outside of the liposomes and we initiated the transport by adding 2 mM EDTA to the outside to chelate all external $Co^{2+}$ (Fig. 4e). We observed rapid quench of pyranine fluorescence and the quench reached ~74% at the end of 400 s (Fig. 4f, g). As controls, there was almost no change in fluorescence when no $Co^{2+}$ was included in the system, and a small change in fluorescence when using liposomes with no TsFpn (Fig. 4f, g). These results indicate that $Co^{2+}$ transport by TsFpn is coupled to $H^+$ transport in the opposite direction. Third, we estimated the stoichiometry of the coupled $H^+$ and $Co^{2+}$ movement by examining if metal ion transport in TsFpn is influenced by a membrane potential (Fig. 4j and Methods section). We clamped the membrane potential at $-120$ mV, 0 mV, and $+120$ mV by using a proper $K^+$ gradient in combination with valinomycin (Methods section), and we found that the rate of $Co^{2+}$ uptake is not significantly different in all three membrane potentials (Fig. 4k, n). These results indicate that metal ion transport in TsFpn is electroneutral and thus transport of each divalent metal ion is accompanied by transport of two $H^+$ in the opposite

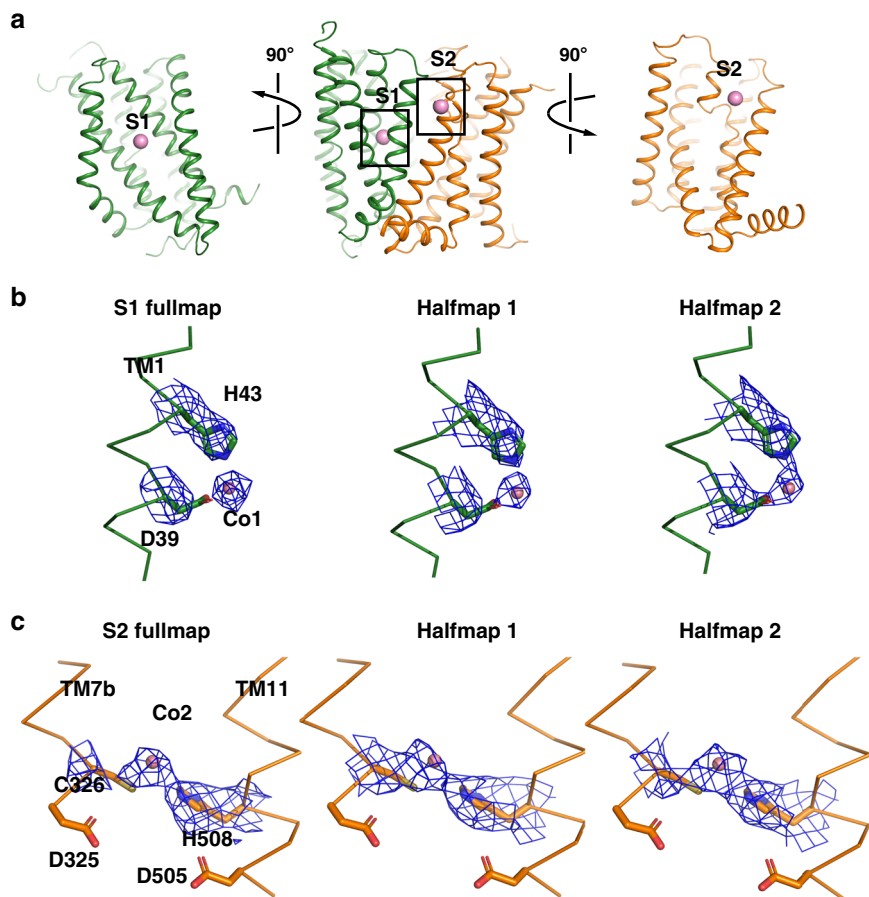

**Fig. 3 Two ion binding sites in TsFpn. a** TsFpn in cartoon representation is shown in three orientations with the locations of S1 and S2 marked by the bound $Co^{2+}$ (pink spheres). **b** Density map around S1 is shown as blue mesh. Part of the TM1 is shown as trace and the side chains of Asp39 and His43 are shown in stick. **c** Density map around S2 is shown as blue mesh. Part of the TM7 and 11 are shown as trace and the side chains of Asp325, Cys326, Asp505, and His508 are shown as sticks.

direction. Taken together, these results are consistent with the hypothesis that TsFpn is an electroneutral $H^+/Fe^{2+}$ antiporter.

**Mutational study of metal ion binding sites**. As a second test to validate the metal binding sites, we made mutations to the S1 and S2 binding sites and measured ion binding and transport. Two double mutations were made: the S1 mutation, i.e., Asp39Ala-His43Ala, and the S2 mutation, i.e., Cys326Ala-His508Ala. Both the S1 and S2 mutations can be purified and are stable after purification (Supplementary Fig. 1e, f).

The S1 mutation has a slightly reduced rate of $Co^{2+}$ transport $(0.10 \pm 0.0015 \, min^{-1})$ to that of the wild type TsFpn in symmetrical pH 7.5 (Fig. 4h and Supplementary Table 2). The modest change in $Co^{2+}$ transport suggests that when the S1 site is mutated, the S2 site can still mediate metal ion transport. However, the S1 mutation has a larger impact in $H^+$ transport. The rate of $Co^{2+}$ transport in the S1 mutant increases modestly to $0.17 \pm 0.0072 \, min^{-1}$ (Fig. 4h, i) under a pH gradient (pH 7.5 inside and pH 8.5 outside). The change in the rate of uptake (~1.7 fold) in the S1 mutation from symmetrical pH to a $\Delta pH = 1$ gradient is significantly smaller than the change in the wild type transporter (~4.3 fold, Fig. 4i). This result suggests that the S1 mutation is less sensitive to a pH gradient and thus the coupled transport of $H^+$ and $Co^{2+}$ may have been compromised. Consistent with this conclusion, $Co^{2+}$ transport in the S1 mutation becomes electrogenic. At $-120 \, mV$ membrane potential, the rate of transport is ~2.4 fold faster than that at 0 mV, and the increase is significantly larger than that of the wild

type (~1.05 fold, Fig. 4l, n). The direction of change is consistent with transport of $<2 \, H^+$ for each $Co^{2+}$. We also measured the rate of transport at $+120 \, mV$, and as expected, the rate becomes slower. Although we cannot determine the exact stoichiometry of $H^+$ and $Co^{2+}$ in the S1 mutation due to the qualitative nature of the fluorescence quench assay, these results are consistent with the conclusion that the coupled $H^+$ transport is reduced.

The S2 mutation has a significantly slower rate of metal ion uptake $(0.067 \pm 0.0057 \, min^{-1})$ than that of the wild type under symmetrical pH (Fig. 4h and Supplementary Table 2). This indicates that the S2 site is more crucial than the S1 site for metal ion transport. The S2 mutant is also less sensitive to a pH gradient and has a modest increase in the rate of $Co^{2+}$ transport $(0.12 \pm 0.0014 \, min^{-1})$ under a pH gradient (pH 7.5 inside and pH 8.5 outside). The change in the rate of uptake (~1.7 fold) in the S2 mutation from symmetrical pH to a $\Delta pH = 1$ gradient is also significantly smaller than the change in the wild type transporter (Fig. 4i). Also similar to the S1 mutant, the rate of $Co^{2+}$ transport is enhanced under $-120 \, mV$ membrane potential and reduced under $+120 \, mV$ (Fig. 4m, n). Combined, these results suggest that the S2 site has a more prominent role in both metal ion and $H^+$.

We also monitored $H^+$ transport in the S1 and S2 mutants using the pyranine assay. The rate of $H^+$ transport is slower in both the S1 and S2 mutants (Fig. 4f, g), and the S2 mutation has a larger impact on the rate which seems to mirror the effect observed in the $Co^{2+}$ transport assay. Combined, results from the functional studies on the S1 and S2 mutations are consistent with

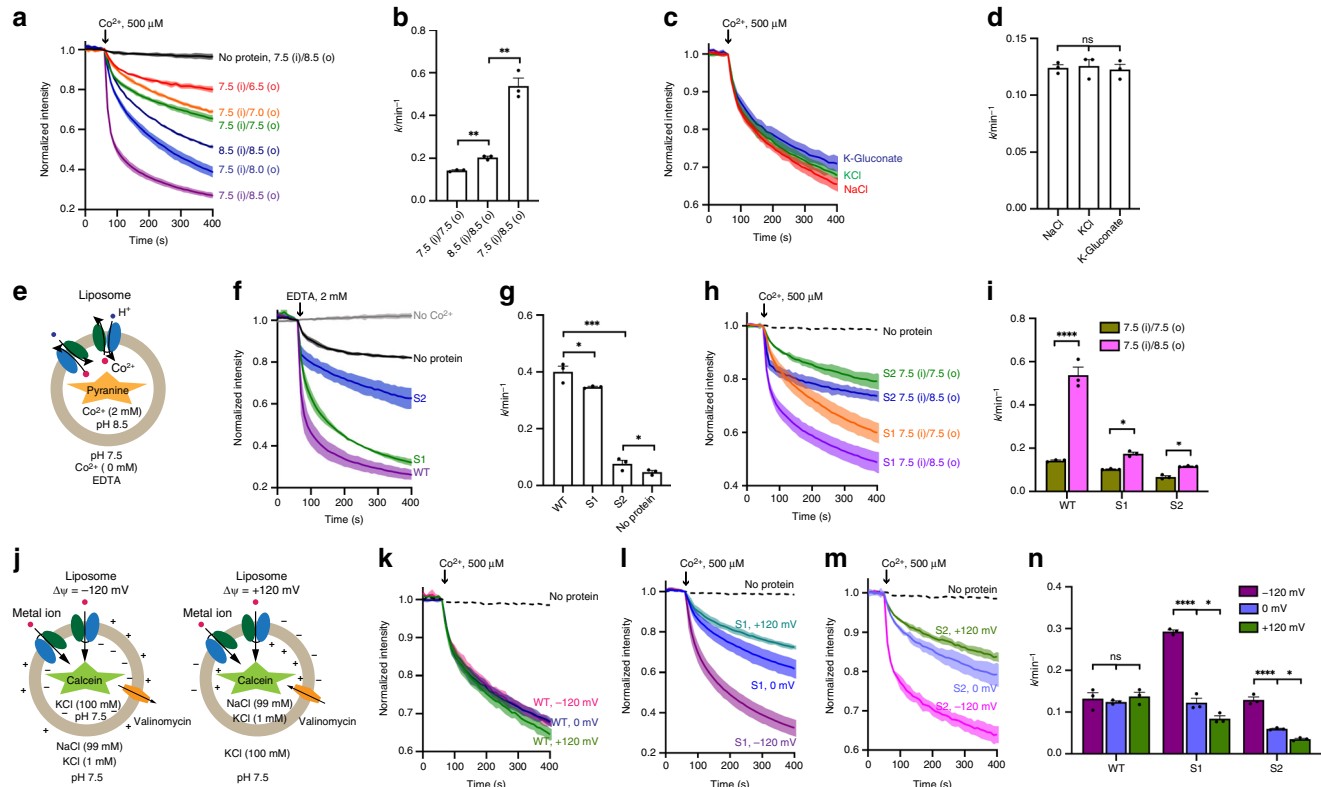

**Fig. 4 TsFpn is a H$^+$/Fe$^{2+}$ antiporter. a** Quench of calcein fluorescence in different external and internal pH conditions. **b** Initial rate constants of Co$^{2+}$ uptake in different pH gradient. Two-tailed Student's $t$ test: **$p < 0.01$. **c** Quench of calcein fluorescence under symmetrical KCl, symmetrical NaCl, and symmetrical K-Gluconate. **d** Initial rate constants of Co$^{2+}$ uptake in buffer with different ions. One-way ANOVA: no significant difference ($p = 0.89$). **e** Schematic view of a flux assay that monitors H$^+$ influx when Co$^{2+}$ flows out of the liposomes. **f** Quench of pyranine fluorescence for liposomes that with the wild-type TsFpn (WT, purple), S1 mutant (S1, green), S2 mutant (S2, blue), and no protein (black). **g** Initial rate constants of H$^+$ uptake of WT and mutants. Two-tailed Student's $t$ test: *$p < 0.05$; ***$p < 0.001$. **h** Quench of calcein fluorescence in liposomes with the S1 mutant in symmetrical pH 7.5 (orange) and pH 7.5 (i)/ 8.5 (o) (purple), and with the S2 mutant in symmetrical pH 7.5 (green) and pH 7.5 (i)/ 8.5 (o) (blue). **i** Initial rate constants of Co$^{2+}$ uptake of WT and mutants in different pH gradient. Two-way ANOVA: among different proteins, $p < 0.0001$; between different pH gradients, $p < 0.0001$; interaction, $p < 0.0001$; the pH 7.5 (i)/ 8.5 (o) differs from the symmetrical pH 7.5 in WT ($p < 0.0001$), S1 ($p = 0.022$), and S2 ($p = 0.041$). **j** Schematic view of TsFpn-containing proteoliposomes with −120 mV membrane potentials (left) and +120 mV (right). **k-m** Quench of calcein fluorescence over time measured in three membrane potentials for the wild type (**k**), the S1 (**l**) and S2 (**m**) mutant. **n** Initial rate constants of Co$^{2+}$ uptake of WT and mutants at −120 or +120 mV. Two-way ANOVA: among different proteins, $p < 0.0001$; among different membrane potentials, $p < 0.0001$; interaction, $p < 0.0001$; within each group, statistical significances are indicated: ns not significant; *$p < 0.05$; ****$p < 0.0001$.

structural assignment of S1 and S2 as metal ion binding sites, and reinforced the conclusion that TsFpn is a H$^+$/Fe$^{2+}$ antiporter.

To estimate how the two sites contribute to metal ion binding, we measured Co$^{2+}$ binding to the S1 and S2 mutations by ITC. The S1 mutation binds to Co$^{2+}$ with a $K_d$ of 266.3 ± 23.8 µM, and the S2 mutation binds to Co$^{2+}$ with a $K_d$ of 162.4 ± 16.0 µM (Supplementary Fig. 2h, i). When both sites are mutated (Asp39Ala/His43Ala/Cys326Ala/His508Ala, Supplementary Fig. 1g), the binding affinity becomes 616.0 ± 44.9 µM (Supplementary Fig. 2j). These results suggest that the S1 site has slightly higher affinity for Co$^{2+}$ than the S2 site, although it is difficult to correlate the affinity of a site with their roles in transport of the ions. More accurate assays for both ion binding and transport are required to unravel the mechanism of ion transport in Fpn.

**Inhibition by hepcidin.** To understand how hepcidin inhibits Fpn, we measured inhibition of TsFpn by human hepcidin, which is almost identical to tarsier hepcidin with two conservative replacements at position 8 (Met in tarsier versus Ile in human) and position 16 (Lys in tarsier versus Arg in human). Addition of hepcidin (20 µM) to the external side of the liposomes reduces the rate of transport (100 µM Co$^{2+}$) from 0.11 ± 0.0057 to 0.053 ±

0.0067 min$^{-1}$. When both hepcidin and 11F9 Fab were added to the external side, the rate of transport was further reduced to 0.032 ± 0.0039 min$^{-1}$ (Fig. 5a–c and Supplementary Table 2). The additive effect in inhibition is consistent with the conclusion that TsFpn assumes both orientations when reconstituted into liposomes. Inclusion of hepcidin on both inside and outside of the liposomes also has an additive effect and reduces the rate of transport to 0.043 ± 0.0002 min$^{-1}$. These results suggest that hepcidin does not fully inhibit the transport activity. The partial inhibition is not due to a mismatch between human hepcidin and TsFpn, because a similar partial inhibition was also found when we measured inhibition of human Fpn by human hepcidin (Fig. 5d, e). This phenomenon was seen in a previous study in which hepcidin (10 µM) inhibits ~50% of iron export in a cell-based assay[3].

We proceeded to solve the structure of TsFpn in the presence of 250 µM of hepcidin to an overall resolution of 3.4 Å (Fig. 5f and Supplementary Figs. 3f–j and 4b). The density map is of sufficient quality to resolve and build the structure of TsFpn. The density map has a prominent non-protein density located between the N- and C-domains on the extracellular side which is not present in the previous apo-TsFpn structure. The new density interacts with both the N- and C-domains, and reaches

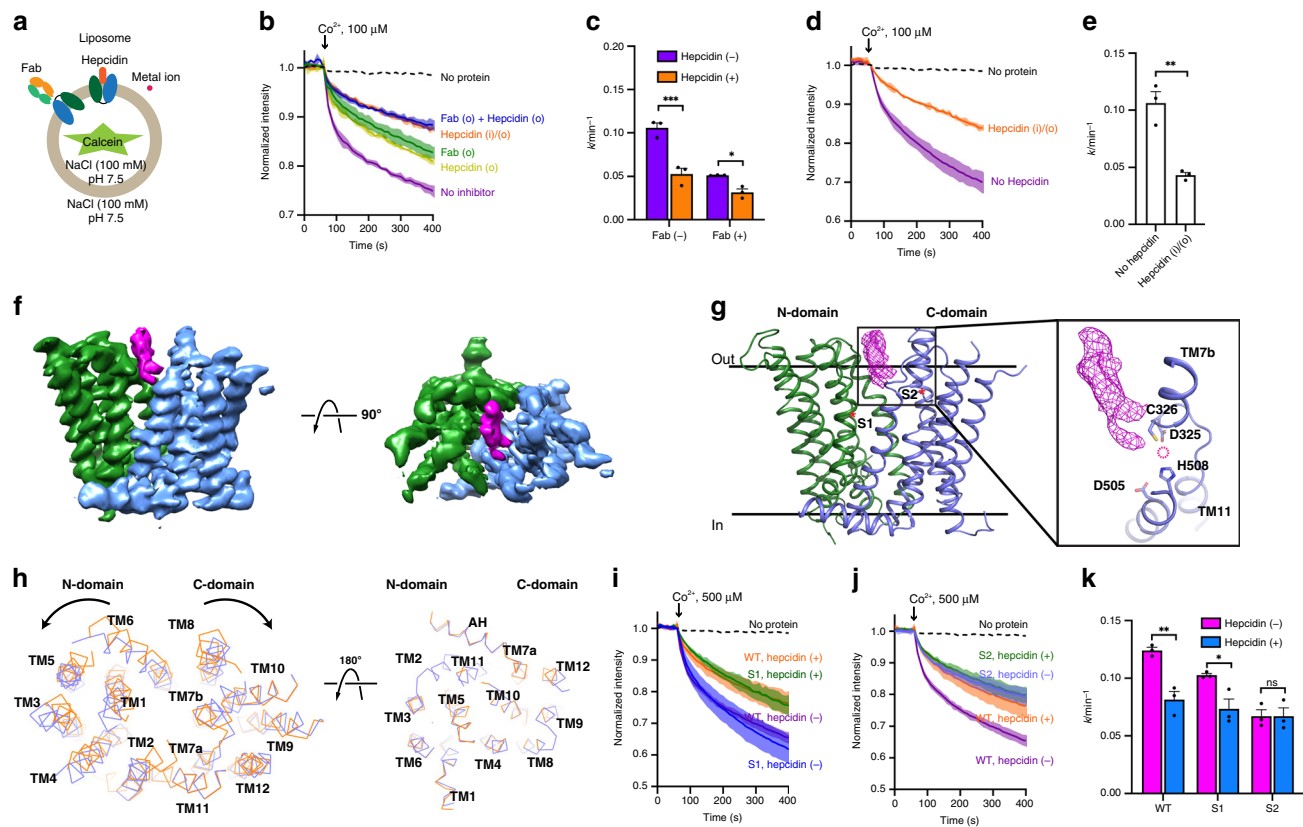

**Fig. 5 Hepcidin inhibition of TsFpn. a** Schematic view of orientation-specific binding of hepcidin and Fab to TsFpn in proteoliposomes. **b** Fluorescence quench mediated by TsFpn with no hepcidin (purple), with hepcidin on the outside (light green), with Fab on the outside (green), with hepcidin on both in- and outside (orange), and with both Fab and hepcidin on the outside (blue). **c** Initial rate constants of $Co^{2+}$ uptake in the presence of different inhibitors. Two-way ANOVA: interaction, $p = 0.0083$. Within each group, statistical significances are indicated: $*p < 0.05$; $***p < 0.001$. **d** Fluorescence quench mediated by human Fpn with no hepcidin (purple), with hepcidin on both inside and outside (orange). **e** Initial rate constants of $Co^{2+}$ uptake. Two-tailed Student's $t$ test: $**p < 0.01$. **f** Cryo-EM map of TsFpn (green and blue) in complex with hepcidin (magenta) in two orientations. The Fab density is not shown. **g** Structure model of TsFpn in the presence of hepcidin. Hepcidin density is shown as magenta mesh. Positions of S1 and S2 are labeled as red stars. Inset: close-up view of hepcidin density and the S2 site. $Co^{2+}$ position is marked as a dotted circle. **h** Overlay of TsFpn structure with (blue) and without hepcidin (orange), viewed from the extracellular (left panel) and intracellular side (right panel). The relative movement of N- and C-domains is marked with arrows. **i, j** Fluorescence quench mediated by the S1 (**i**) and S2 (**j**) mutations in the presence or absence of hepcidin. **k** Initial rate constants of $Co^{2+}$ uptake of WT and mutants with or without hepcidin. Two-way ANOVA: interaction, $p = 0.015$; within each group, statistical significances are indicated: ns not significant; $*p < 0.05$; $**p < 0.01$.

deep into the transporter to the S2 site (Fig. 5g). The new density is not of sufficient quality for building a hepcidin structure, however, in the presence of hepcidin, the extracellular side of the N- and C-domains of TsFpn moves away from each other compared to that of apo-TsFpn, while the intracellular side of the N- and C-domains remains well-aligned (Fig. 5h). Since the hepcidin density reaches the S2 site but not the S1 site, we examined hepcidin inhibition on the two mutations. Consistent with the structure, the S2 mutation is insensitive to hepcidin while the S1 mutation remains inhibited by hepcidin (Fig. 5i–k). Mutations to Cys326, which is part of the S2 site, are known to cause ferroportin diseases by affecting hepcidin binding and it was proposed that Cys326 could form a disulfide bridge with Cys7 on hepcidin[36,37]. The current structure cannot resolve if there is a disulfide bond. In summary, the structure of TsFpn-hepcidin complex shows that hepcidin binds to an outward-facing Fpn and reaches one of the metal ion binding sites. The structure suggests that hepcidin could inhibit $Fe^{2+}$ transport by obstructing the outward-facing to inward-facing conformational changes or by interfering with binding of ions, or both.

## Discussion

In summary, we solved the structure of TsFpn in an outward-facing conformation with and without hepcidin. We identified two potential metal ion binding sites S1 and S2 and we demonstrated that TsFpn is an electroneutral $2H^+/Fe^{2+}$ antiporter. The electroneutral export of $Fe^{2+}$ is likely critical to the physiological functions of Fpn, because cells have a negative resting membrane potential that makes unidirectional export of cations energetically unfavorable and the coupled import of $H^+$ can facilitate the iron export. The structures also show that hepcidin binds to the outward-facing conformation of Fpn and reaches the S2 site.

Given the high structural similarity between the outward-facing conformations of TsFpn and BbFpn, we generated a model of TsFpn in the inward-facing conformation by aligning its N- and C-domains separately on their equivalent domains in the inward-facing BbFpn structure (PDB ID 5AYO; Supplementary Fig. 8a). The transmembrane domains of the two structures align reasonably well with a root mean square distance of 2.8 Å for the backbone atoms of the N-domain and 1.6 Å for these of the C-domain. Both the S1 and S2 sites are solvent accessible from the intracellular side in the inward-facing model of TsFpn

(Supplementary Fig. 8b, c), suggesting that a canonical rocker-switch type motion of the N- and C-domains could achieve alternating access to the substrates (Supplementary Fig. 9).

The structure does not define binding sites for H$^+$. Since both the S1 and S2 sites contain a histidine that could mediate H$^+$ binding and transport, we speculate that the H$^+$ and Fe$^{2+}$ share the binding sites but do not occupy them simultaneously. If so, then we could sketch a transport scheme to provide a tentative overview of how export of one Fe$^{2+}$ is coupled to uptake of two H$^+$ via the S1 and S2 sites (Supplementary Fig. 9). In this model, each transport cycle exports one Fe$^{2+}$ that binds sequentially to the S1 and S2 sites and imports two H$^+$ that occupy both S1 and S2. Further studies are required to resolve the transport mechanism.

## Methods

### Cloning, expression, and purification of TsFpn
The Fpn gene (accession number XP_008060857) from *Carlito syrichta* (*Tarsius syrichta*, Philippine tarsier) was codon-optimized and cloned into a pFastBac dual vector[38] for production of baculovirus by the Bac-to-Bac method (Invitrogen). High Five Cells (Thermofisher) at a density of ~$3 \times 10^6$ cells/ml were infected with baculovirus and grown at 27 °C for 60–70 h before harvesting. Cell membranes were prepared following a previous protocol[38] and frozen in liquid nitrogen.

Purified membranes were thawed and homogenized in 20 mM HEPES, pH 7.5, 150 mM NaCl and 2 mM β-mercaptoethanol, and then solubilized with 1% (w/v) Lauryl Maltose Neopentyl Glycol (LMNG, Anatrace) at 4 °C for 2 h. After centrifugation (55,000 × $g$, 45 min, 4 °C), TsFpn was purified from the supernatant using a cobalt-based affinity resin (Talon, Clontech) and the His$_6$-tag was cleaved by TEV protease at room temperature for 1 h. TsFpn was then concentrated to 3–6 mg/ml (Amicon 50 kDa cutoff, Millipore) and loaded onto a size-exclusion column (SRT-3C SEC-300, Sepax Technologies, Inc.) equilibrated with 20 mM HEPES, pH 7.5, 150 mM NaCl, 1 mM (w/v) n-dodecyl-β-D-maltoside (DDM, Anatrace).

Mutations to TsFpn were generated using the QuikChange method (Stratagene) and the entire cDNA was sequenced to verify the mutation. The primers information is provided in Supplementary Table 3. TsFpn mutants and wild type HsFpn were expressed and purified following the same protocol as wild type TsFpn.

### Generation of monoclonal antibodies and Fab fragments
Monoclonal antibodies against the TsFpn (IgG2b, κ) were raised using standard methods (Monoclonal Core, Vaccine and Gene Therapy Institute, Oregon Health & Science University). High affinity and specificity of the antibodies for properly folded TsFpn was assayed by ELISA and western blot (no binding). Three out of twenty antibodies were selected for large scale production. Fab fragments were generated by papain cleavage of whole antibody at a final concentration of 1 mg/mL for 2 hours at 37 °C in 50 mM Phosphate buffer saline, pH 7.0, 1 mM EDTA, 10 mM cysteine and 1:50 w-w papain:antibody. Digestion was quenched using 30 mM iodoacetamide at 25 °C for 10 min. Fab was purified by anion exchange using a Q Sepharose (GE Healthcare) column in 10 mM Tris, pH 8.0 and a NaCl gradient elution. TsFpn-Fab complexes were further verified by size-exclusion chromatography (shift in elution volume and SDS-PAGE) and 11F9 was selected for structural studies.

### TsFpn-11f9(Fab) complex
Purified TsFpn was mixed with the 11F9 Fab at 1:1.1 molar ratio and incubated 30 min on ice. The complex was then concentrated to 3–6 mg/ml (Amicon 100 kDa cutoff, Millipore) and loaded onto a size-exclusion column equilibrated with 20 mM HEPES, pH 7.5, 150 mM NaCl, 1 mM n-dodecyl-β-D-maltoside (DDM, Anatrace). The TsFpn-Fab complex was used in the ITC measurement of Co$^{2+}$ binding and in nanodisc reconstitution for cryo-EM grid preparations.

### Nanodisc reconstitution
MSP1D1 was expressed and purified following an established protocol[39]. For lipid preparation, 1-palmitoyl-2-oleoyl-sn-glycero-3-phospho-(1'-rac)-choline (POPC, Avanti Polar Lipids), 1-palmitoyl-2-oleoyl-sn-glycero-3-phospho-(1'-rac)-ethanolamine (POPE, Avanti Polar Lipids) and 1-palmitoyl-2-oleoyl-sn-glycero-3-phospho-(1'-rac)-glycerol (POPG, Avanti Polar Lipids) were mixed at a molar ratio of 3:1:1, dried under Argon and resuspended with 14 mM DDM[40]. For nanodisc reconstitution, TsFpn, Fab of 11F9, MSP1D1 and lipid mixture were mixed at a molar ratio of 1:(1.1):(2.5):(62.5) and incubated on ice for 1 h. Detergents were removed by incubation with Biobeads SM2 (Bio-Rad) overnight at 4 °C. The protein lipid mixture was loaded onto a size-exclusion column equilibrated with 20 mM HEPES, pH 7.5, 150 mM NaCl. The purified nanodisc elutes at 13.6 ml and was concentrated to 13 mg/ml and incubated with 10 mM CoCl$_2$ or 250 μM human hepcidin (Sigma aldrich) for 30 min before cryo-EM grid preparation.

### Cryo-EM sample preparation and data collection
The cryo grids were prepared using Thermo Fisher Vitrobot Mark IV. The Quantifoil R1.2/1.3 Cu grids were glow-discharged with air for 15 s at 10 mM in a Plasma Cleaner (PELCO Easi-Glow$^{TM}$). Aliquots of 3.5 μl purified TsFpn-Fab in nanodisc were applied to glow-discharged grids. After being blotted with filter paper (Ted Pella, Inc.) for 4.0 s, the grids were plunged into liquid ethane cooled with liquid nitrogen. A total of 1838 micrograph stacks were collected for TsFpn-Fab and 1381 for TsFpn-Fab-hepcidin with SerialEM[41] on a Titan Krios at 300 kV equipped with a K2 Summit direct electron detector (Gatan), a Quantum energy filter (Gatan) and a Cs corrector (Thermo Fisher), at a nominal magnification of 105,000× and defocus values from −2.0 μm to −1.2 μm. Each stack was exposed in the super-resolution mode for 5.6 s with an exposing time of 0.175 s per frame, resulting in 32 frames per stack. The total dose rate was about 50 e$^-$/Å$^2$ for each stack. The stacks were motion corrected with MotionCor2[42] and binned 2 fold, resulting in a pixel size of 1.114 Å/pixel. In the meantime, dose weighting was performed[43]. The defocus values were estimated with Gctf[44].

### Cryo-EM data processing
For the TsFpn-Fab dataset, a total of 1,246,999 particles were automatically picked with RELION 2.1[45–47]. After 2D classification, a total of 946,473 particles were selected and subject to a global angular search 3D classification with one class and 40 iterations. The outputs of the 35th–40th iterations were subjected to local angular search 3D classification with four classes separately. A total of 571,511 particles were selected by combining the good classes of the local angular search 3D classification. After handedness correction, a skip-align classification procedure was performed to further classify good particles, yielding a total of 215,752 particles, which resulted into a reconstruction with an overall resolution of 3.1 Å after 3D auto-refinement with an adapted mask. The resolution of the map was further improved to 3.0 Å after Bayesian polishing[48].

For the TsFpn-Fab-hepcidin dataset, a total of 2,279,202 particles were automatically picked with RELION 2.1[45–47]. After 2D classification, a total of 540,037 particles were selected and subject to 5 rounds of a multi-reference global search 3D classification with 4 class and 40 iterations. The outputs of the 35th–40th iterations of the last round containing 313,890 particles were subjected to local angular search 3D classification with four classes separately. A total of 214,136 particles were selected by combining the good classes of the local angular search 3D classification. After handedness correction and Bayesian polishing, 3D auto refinement with an adapted mask yielded a reconstruction with an overall resolution of 3.2 Å. After postprocessing, the resolution of the final map is 3.4 Å, probably because this dataset has slight orientation preference.

All 2D classification, 3D classification, and 3D auto-refinement were performed with RELION 2.1 or RELION 3.0. Resolutions were estimated with the gold-standard Fourier shell correlation 0.143 criterion[49] with high-resolution noise substitution[50].

### Model building and refinement
For de novo model building of TsFpn-11F9 complex, a ploy-Alanine model was first manually built into the 3.0 Å density map in COOT[51] and side chains were added next. Structure refinements were carried out by PHENIX in real space with secondary structure and geometry restraints[52]. TsFpn-11F9-hepcidin structure was built based on the apo-TsFpn structure and the model was adjusted manually into the density map in COOT, and refined by PHENIX. The EMRinger Score was calculated as described[53]. All structural figures were generated in Pymol (https://pymol.org/2/) and Chimera[54].

### Proteoliposome preparation
POPE and POPG (Avanti Polar Lipids) was mixed at 3:1 molar ratio, dried under Argon and vacuumed overnight to remove chloroform. The lipid was resuspended in the reconstitution buffer (20 mM HEPES, pH 7.5, 100 mM NaCl) to a final concentration of 10 mg/ml, sonicated to transparency and incubated with 40 mM n-decyl-β-D-maltoside (DM, Anatrace) for 2 h at room temperature under gentle agitation. Wild type or mutant TsFpn was added at 1:100 (w/w, protein:lipid) ratio. The detergent was removed by dialysis at 4 °C against the reconstitution buffer. Dialysis buffer was changed once a day and the liposomes were harvested after 4 days, aliquoted, and frozen at −80 °C.

### Divalent metal ion transport assay
Liposomes were mixed with 250 μM calcein and frozen-thawed three times. After the liposomes were extruded to homogeneity with 400 nm filter (NanoSizer$^{TM}$ Extruder, T&T Scientific Corporation), free calcein was removed through a desalting column (PD-10, GE Healthcare) equilibrated with the dialysis buffer. Calcein fluorescence was monitored in a quartz cuvette at 37 °C. Fluorescence was monitored in a FluoroMax-4 spectrofluorometer (HORIBA) with 494 nm excitation and 513 nm emission at 10 s intervals. The transport was initiated by the addition of 0.5 mM CoCl$_2$ or 100 μM fresh prepared FeSO$_4$. When Fe$^{2+}$ is used, 1 mM vitamin C was added in the external solution.

In experiments when internal solution needs to be replaced, liposomes were centrifuged at 47,000 × $g$ for 30 min and resuspended in a desired internal solution. A fluorescent dye was then loaded into the liposomes by the same freeze-thaw processes and free dye was removed by a desalting column. The concentration of valinomycin and hepcidin is 1 μM and 20 μM when used, respectively.

**Pyranine assay**. Liposomes were centrifuged at $47,000 \times g$ for 30 min and resuspended in inside buffer (5 mM Tris, pH 8.5, 100 mM NaCl). Liposomes were mixed with 250 μM pyranine and 2 mM $CoCl_2$ and underwent three freeze-thaw cycles. After the liposomes were extruded to homogeneity with 400 nm filter (NanoSizer™ Extruder, T&T Scientific Corporation), free dye was removed through a desalting column (PD-10, GE Healthcare) equilibrated with the outside buffer (5 mM HEPES, pH 7.5, 100 mM NaCl, 2 mM $CoCl_2$). Pyranine fluorescence was monitored in a quartz cuvette at 37 °C in a FluoroMax-4 spectrofluorometer (HORIBA) with 460 nm excitation and 510 nm emission at 10 s internals. The transport was initiated by the addition of 2 mM EDTA.

**Transport data analysis and statistics**. Fluorescence quench has a fast initial phase and a slower second phase. We focused on the initial fast phase. The rate of uptake is estimated by fitting the first 60 s of data points with a single exponential decay function and the rate constants were plotted in bar graphs. Two-way analysis of variance (ANOVA) was used where appropriate and the follow-up multiple comparison within groups was carried out with Holm-Sidak test. For transport conditions with one variable, one-way ANOVA was used to test for differences among multiple groups. Two-tailed Student's $t$ test was performed for pairwise comparison. All statistical analyses were performed in GraphPad Prism 8.2.1.

In an enzyme with a canonical Michaelis–Menten kinetics, $K_M = \frac{k_r + k_{cat}}{k_f} > K_d = \frac{k_r}{k_f}$, in which $k_f$ and $k_r$ are the rates of binding and unbinding of a substrate to the enzyme and $k_{cat}$ is the rate of substrate conversion. For a transporter, both $K_M$ and $K_d$ are more complicated because more kinetic steps are involved. After a substrate binds to a transporter, the transporter undergoes a structural change to translocate the substrate to the other side of the membrane, followed by substrate dissociation and then another structural change to recovery of the transporter to the initial conformation. The affinity of a substrate to the outward-facing and inward-facing conformations could be different, and substrate affinity could be further complicated by additional conformations of the transporter and the required co-transporting substrates.

**Isothermal titration calorimetry**. Protein samples were purified as described above and concentrated to around 50–100 μM (3–6 mg/ml). TsFpn was in the ITC buffer that contains 20 mM HEPES pH 7.5, 150 mM NaCl, 1 mM DDM. The ITC measurements were performed with a Nano ITC microcalorimeter (TA Instruments) at 25 °C. $CoCl_2$ stock at 5 mM was prepared in the same ITC buffer injected 25 times (1.01 μl for injection 1 and 2.02 μl for injections 2–25), with 175 s intervals between injections. The background data obtained from injecting $Co^{2+}$ into the ITC buffer were subtracted before the data analysis. The data were fitted using NanoAnalyze v3.11.0 (TA Instruments). Heat releases from lower ligand to protein ratios were not determined because of the modest affinity.

**Reporting summary**. Further information on research design is available in the Nature Research Reporting Summary linked to this article.

## Data availability
Data supporting the findings of this manuscript are available from the corresponding authors upon reasonable request. A reporting summary for this Article is available as a Supplementary Information file. The atomic coordinates of TsFpn-Fab complex at Co-bound and hepcidin-bound state have been deposited in the PDB (http://www.rcsb.org) under the accession codes 6VYH https://doi.org/10.2210/pdb6VYH/pdb and 6WIK https://doi.org/10.2210/pdb6WIK/pdb, respectively. The corresponding electron microscopy maps have been deposited in the Electron Microscopy Data Bank under the accession codes https://www.ebi.ac.uk/pdbe/entry/emdb/EMD-21460 and https://www.ebi.ac.uk/pdbe/entry/emdb/EMD-21684, respectively. Source data are provided with this paper.

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

## Acknowledgements

This work was supported by grants from NIH (DK122784, HL086392, and GM098878 to M.Z.), Cancer Prevention and Research Institute of Texas (R1223 to M.Z.). Ara Parse-ghian Medical Research Foundation (to N.Y.). N.Y. is supported by the Shirley M. Tilghman endowed professorship from Princeton University. We thank Paul Shao for technical support during EM image acquisition. We acknowledge the use of Princeton's Imaging and Analysis Center, which is partially supported by the Princeton Center for Complex Materials, and the National Science Foundation (NSF)-MRSEC program (DMR-1420541).

## Author contributions

M.Z., Z.R., Y.P., and J.S. conceived the project. Y.P., Z.R., S.G., J.S., L.W., Y.Y., H.Z., Z.X., P.B., X.F., and A.L. conducted experiments. S.G. led the effort of cryo-EM grid preparation and data collection and was assisted by other authors. Y.P., Z.R., S.G., J.S., Z.X., P.B., A.L., N.Y., and M.Z. analyzed data. Z.R., J.S., Y.P., and M.Z. wrote the initial draft and all authors participated in revising the manuscript.

## Competing interests

The authors declare no competing interests.
