## [Peer Review File · Nature Communications]

REVIEWERS' COMMENTS

Reviewer #1 (Remarks to the Author):

The authors have adequately addressed all my previous points, and also those of reviewer#3

I have just one one minor suggestion, to avoid confusion:

The authors have now included an extensive discussion on ref 23 in order to give a potential explanation for the K_M/K_d "anomaly". I find this reference out of place. For the experiments in ref 23 on the HXT proteins, efflux plays a crucial role to explain the anomaly. In the case of the authors' experiments on Herceptin, efflux cannot play any role (as they explained in the previous rebuttal!), because all transported ions are sequestered by the dye molecule, making the reaction unidirectional. My suggestion is to remove reference 23, and state instead:

"It has not escaped our notice that K_M is ~ 20 fold smaller than K_d . For a simple enzymatic reaction composed of substrate binding and then conversion to a product, K_M is always larger than K_d (Methods).

The reversal of K_M and K_d is an indication that Fpn undergoes multiple kinetic states to transport metal

ions. However, we do not have enough data to lay out the full kinetic mechanism of transport. In addition, K_d was measured on TsFpn solubilized in detergent while K_M on proteoliposomes. More experiments are needed to address the mechanism of transport. Interestingly, a

similar phenomenon was reported for a mitochondrial iron exporter in which the K_M is $4 \mu\text{M}$ and K_d is $450 \mu\text{M}$.

REVIEWERS' COMMENTS

Reviewer #1 (Remarks to the Author):

The authors have adequately addressed all my previous points, and also those of reviewer#3

I have just one one minor suggestion, to avoid confusion:

The authors have now included an extensive discussion on ref 23 in order to give a potential explanation for the K_m/K_d "anomaly". I find this reference out of place. For the experiments in ref 23 on the HXT proteins, efflux plays a crucial role to explain the anomaly. In the case of the authors' experiments on Herceptin, efflux cannot play any role (as they explained in the previous rebuttal!), because all transported ions are sequestered by the dye molecule, making the reaction unidirectional. My suggestion is to remove reference 23, and state instead:

"It has not escaped our notice that K_M is ~ 20 fold smaller than K_d . For a simple enzymatic reaction composed of substrate binding and then conversion to a product, K_M is always larger than K_d (Methods). The reversal of K_M and K_d is an indication that Fpn undergoes multiple kinetic states to transport metal ions. *However, we do not have enough data to lay out the full kinetic mechanism of transport* . In addition, K_d was measured on TsFpn solubilized in detergent while K_M on proteoliposomes. More experiments are needed to address the mechanism of transport. Interestingly, a similar phenomenon was reported for a mitochondrial iron exporter in which the K_M is 4 μM and K_d is 450 μM ²⁴.

We have revised the text following Reviewer 1's suggestion.